# Role of Platelets in Osteoarthritis—Updated Systematic Review and Meta-Analysis on the Role of Platelet-Rich Plasma in Osteoarthritis

**DOI:** 10.3390/cells11071080

**Published:** 2022-03-23

**Authors:** Ewa Tramś, Kamila Malesa, Stanisław Pomianowski, Rafał Kamiński

**Affiliations:** Centre of Postgraduate Medical Education, Department of Orthopaedics and Trauma Surgery, Professor A. Gruca Teaching Hospital, Konarskiego 13, 05-400 Otwock, Poland; ewa.trams@gmail.com (E.T.); kama.malesa@gmail.com (K.M.); spom@spskgruca.pl (S.P.)

**Keywords:** PRP, platelet-rich plasma, platelets, osteoarthritis, inflammation

## Abstract

Platelets are an essential component of hemostasis, with an increasing role in host inflammatory processes in injured tissues. The reaction between receptors and vascular endothelial cells results in the recruitment of platelets in the immune response pathway. The aim of the present review is to describe the role of platelets in osteoarthritis. Platelets induce secretion of biological substances, many of which are key players in the inflammatory response in osteoarthritis. Molecules involved in cartilage degeneration, or being markers of inflammation in osteoarthritis, are cytokines, such as tumor necrosis factor α (TNFα), interleukins (IL), type II collagen, aggrecan, and metalloproteinases. Surprisingly, platelets may also be used as a treatment modality for osteoarthritis. Multiple randomized controlled trials included in our systematic review and meta-analyses prove the effectiveness of platelet-rich plasma (PRP) as a minimally invasive method of pain alleviation in osteoarthritis treatment.

## 1. Osteoarthritis

Osteoarthritis (OA) is a degenerative joint disease that affects between 10% and 18% of middle-aged persons, and up to 44% of older populations [1,2,3]. Although OA was previously deemed to be mechanically activated wear of the cartilage, it is now well established as a low-grade inflammatory disease of the entire joint. OA may be characterized as primary, if there is absence of an antecedent and underlying cause, or secondary, if there is an underlying cause or significant triggering event such as prior trauma [3,4,5].

The main symptoms of OA include pain, stiffness, and loss of function and its sequelae that result in muscle weakness, poor balance, and joint deformity and instability [3,6]. Factors influencing a higher risk and severity of OA are significantly related to older age, female gender, joint injury, anatomic factors, heavy work activities, some professional sport activities, obesity, and genetics [3,4,7]. Additionally, smoking, bone density, and diet remain a focus of OA research [8].

The diagnosis of osteoarthritis is undertaken in symptomatic patients. The radiographic findings of OA can be seen in different modalities, including plain radiographs, ultrasound (US), computed tomography (CT), or magnetic resonance imaging (MRI). Although plain radiograph is the most commonly used modality, MRI has many advantages, including the possibility of assessing joint structures [3,9].

Beneficial treatment of OA may be divided into non-surgical (non-pharmacological and pharmacological) or surgical means [1,10,11]. Non-pharmacological interventions, which include lifestyle modifications such as exercise or weight loss, are the backbone of knee OA management. Irrespective of OA severity, all of these interventions can be used as add-ons to pharmacological treatment. Conventional pharmacological interventions include topical or oral non-steroidal anti-inflammatory drugs (NSAID) and other analgesic drugs, in addition to intra-articular injections with corticosteroids, hyaluronic acid or platelet-rich plasma (PRP). These interventions have proven to be successful, albeit with short-term benefits [12,13].

Surgical treatment plays a significant role when joint-related symptoms persist, despite the use of non-surgical interventions. A growing focus concerns the potential application of experimental methods, which are currently being extensively developed [14,15,16].

## 2. Inflammation—Role in the Pathogenesis of Osteoarthritis

Osteoarthritis is the most common cause of joint degenerative disease, and is a leading cause of disability worldwide; it remains difficult to treat [4]. It has been regarded primarily as a degenerative disease of the cartilage, focusing mainly on acknowledged pathological changes such as cartilage breakdown, the formation of osteophytes, and subchondral bone sclerosis [17]. Although there have been previous studies which indicate that inflammation has a critical role in its pathogenesis, comparatively little attention has been given to the concept of OA pathogenesis involving not only malfunction of cartilage, but also inflammation of the synovial lining [18]. Currently, we no longer view OA as a quintessential degenerative disease resulting from exposure to average bodily wear and tear, but rather as a multifactorial disorder in which low-grade, chronic inflammation has a central role. The involvement of inflammatory components such as cytokines, chemokines, and other inflammatory mediators that are produced by the synovium and chondrocytes, can be easily measured in the synovial fluid of OA patients, and is now well recognized as means of diagnosis [19]. Inflammation is a typical immunological response to pathogens and cellular aberrations that triggers the immune system. Undesired material is primarily recognized by neutrophils, then by specialized phagocytic, megakaryocyte, macrophage cells, and by other cellular mechanisms that are engaged to resolve the pathology and restore homeostasis. Notwithstanding, unresolved chronic inflammation can result in harmful effects. It may result in tissue breakdown and degradation, causing the development of multiple chronic conditions including arthrosclerosis, neurodegenerative diseases, arthritis, inflammatory bowel disease, and cancer. Antecedently, inflammatory arthritis was defined as cellular inflammation described by increased leukocyte numbers in the affected joint tissues and synovial fluid. It was proved that OA is characterized by the presence of a host of proinflammatory mediators, including cytokines and chemokines, which are part of an immune response to joint injury [20]. It remains unclear whether morphological changes that occur in the OA synovium are primary to a systemic immune response or secondary to cartilage degradation and lesions of the subchondral bone [17].

## 3. Osteoarthritis—The Role of Cytokines and Other Inflammatory Mediators

There is a long list of cytokines and chemokines that are detected in OA synovial fluid, such as the following: interleukin (IL) IL-1β, IL-6, IL-7, IL-8, IL-15, IL-17, IL-18, and IL-36; tumor necrosis factor α (TNFα), oncostatin M (OSM), and growth-related oncogene (GRO)-α; chemokine (C-C-motif), ligand 19 (CCL19), macrophage inflammatory protein (MIP)-1β, and transforming growth factor (TGF) α; interferon-induced protein (IP) 10 and monokine induced by interferon (MIG) [20,21,22,23].

Surprisingly, some research has provided evidence to support the hypothesis that inhibition of complement activation in OA by gene deletion or pharmacologic modulation was found to protect mice from surgically induced OA [24,25]. Fragments of matrix proteins including fibronectin, cartilage oligomeric protein (COMP), fibromodulin, proteoglycans, and collagen are released from the damaged matrix. These proteins are able to stimulate the innate immune response and promote the upregulation of degradative pathways via activation of toll-like receptors (TLR) and integrins [20,26,27].

Many of the proinflammatory and catabolic pathways in the OA joint involve activation of the nuclear factor kappa-light-chain-enhancer of activated B cell (NF-kB) family of transcriptional regulators. Other inflammatory mediators include the alarmins (S100 proteins) and the damage-associated molecular patterns (DAMPs). These mediators attract macrophages and control synovitis by enhancing the expression of proteases into the joint. The destruction of joint tissues in OA is mediated by a large number of proteases [28]. Our understanding of the metalloproteinases (MMPs), cysteine proteinases (including cathepsin K), and serine proteinases that are involved in OA is mainly concentrated on possible causes of degradation of cartilage extracellular matrix proteins. Aggrecan is a large proteoglycan that provides much of the resiliency of cartilage. It is degraded in the early stages of OA by members of the ADAMTS (adisintegrin and metalloproteinase with thrombospondin motifs) family, typically referred to as aggrecanases (ADAMTS-4 and -5) [14,28].

Type II collagen, the most abundant collagen in cartilage, is responsible for the tensile strength of cartilage, and is degraded by collagenases. Given the importance of ADAMTS-5 and MMP-13 in OA, development of specific inhibitors to these proteases has been proven as a potentially beneficial, disease-modifying therapy [14]. Thus, we cannot disregard the role of tissue inhibitors of metalloproteinases (TIMPs) that are endogenous inhibitors found in joint tissues and synovial fluid; these may serve as a substitute to synthetic small molecule inhibitors.

MMPs are produced as pro-forms that require proteolytic cleavage in order to be activated. Serin proteases, including HtrA1 and activated protein C, can serve this role and therefore also serve as therapeutic targets in OA research. Cathepsin K is a cysteine proteinase that is expressed by osteoclasts; it can degrade type I collagen in bone but also may degrade type II collagen in cartilage [14,21,28].

Several studies have indicated that there are a number of potential mediators of OA that are not considered proinflammatory mediators, but appear to promote OA by either activating pathways that promote joint tissue destruction, or inhibit the ability of cells to repair the damaged matrix; these are bone morphogenetic proteins (BMPs), fibroblast growth factors (FGF), and Wnts [29,30,31,32].

One further area explored in the novel investigation of OA pathogenesis is epigenetics. There is a growing number of processes, including DNA methylation, histone modifications (acetylation and methylation), microRNAs (miRNA), and long noncoding RNAs, in the epigenetic control of gene transcription. It is not yet known whether epigenetic changes play a key role in the development of OA in humans [33,34,35].

The importance of innate immunity has been associated with the development of synovitis, activation of downstream inflammatory, and catabolic events in articular cartilage that result in progression of OA [19,20].

Therefore, forasmuch as OA might not adhere to the classical presence of cardinal symptoms of inflammation (rubor, calor, dolor, and tumor), there is no strong evidence for the presence of robust adaptive immune reactions; moreover, there is still significant evidence to support a pathogenic role for both local and systemic inflammation in OA development.

## 4. Platelets

Although platelets were traditionally considered essential components of hemostasis, there is growing accedence to the fact that they also play a critical role in the inflammatory and immune responses [36]. The normal range of anucleate discoid platelets is 150,000–400,000/µL, and their main function is to assure hemostasis by preventing loss of blood. Platelets usually circulate for 7 to 10 days; then, they are either used up in hemostasis or undergo apoptosis [37]. Platelets are produced in vast numbers via a multistep process in the bone marrow by megakaryocytes and, as anucleated cells, are incapable of nucleic acid transcription; however, they maintain a meaningful amount of messenger RNA (mRNA), microRNAs and noncoding RNAs (ncRNA). With regard to these features, platelets are able to actively control mRNA translation, perform protein synthesis, and secrete peptides around them which are fundamental features of the complex hemostatic response.

Recent studies have shown increasing evidence that platelets play a significant role in host inflammatory and immune responses [38]. Consequently, antiplatelet medications influence immunity and modify platelet response to inflammation, which result in a significant reduction in mortality rate from either infections or sepsis [39]. In physiological conditions, the role of platelets is related to their reactivity, formed as a result of the reaction between their receptors and vascular endothelial cells in the damaged blood vessel. It is well established that an increase in platelet count is related to the reactivity of megakaryocytes, and to the activity of released pro-platelet cytokines [40,41]. When blood vessels are disturbed by inflammation, platelets are the first cells to be recruited by the vascular endothelium via markers such as selectin and integrin receptors; TLR, complement receptor (CR), cytokines, and chemokines; adenosine diphosphate receptor (ADP), purinoreceptor (PCY); and protease-activated receptors 1 and 4 (PAR-1 and 4). They may be activated and forced to merge with the blood vessel, including polymorphonuclear cells (PMNs), mononuclear cells (MNs), dendritic cells (DC), and T and B cells, usually leading to local inflammation [42,43]. Moreover, the activation of platelets is frequently intensified by P-selectin, which is secreted by the cells of the injured endothelium. P-selectin also stimulates platelets, and activates neutrophils and monocytes towards the activation of endothelial cells [44]. The characteristic features of activated platelets are based on the presence of intracellular adhesion molecules ICAM-1 and monocyte chemoattractant protein-1 (MCP-1), which cause increased adhesion of neutrophils and monocytes. There are many well-known proinflammatory factors that are designated as either recruiters or activators of leukocytes. Their main role is to coordinate the release of clusters of differentiation (CD) 154 (CD154), or CD40 ligands; CXC chemokine ligands (CXCL) CXCL1, CXCL4, CXCL5, CXCL7, and CXCL12; and IL-8 and TGF-β. Platelets induce the secretion of biologically active substances (growth factors and mitogens, coagulation system proteins, cytokines, chemokines, proteolytic enzyme, microRNA, mRNAs, and others), including Weibel–Palade bodies and extracellular vesicles (EV) [45,46,47]; these cause not only a change in the activity of endothelial cells, but also a change in the activity of immune system cells [48,49,50].

The increased exposure of collagen fragments in the damaged endothelium causes inflammation [40,51]. The inflammatory effect of activated platelets is associated with the secretion of platelet-activating factor (PAF) and vascular endothelial growth factor (VEGF); these cause relaxation of the endothelium of blood vessels and increase the inflow of leukocytes and platelets. In addition to this effect, activated platelets can stimulate neutrophils in the neutrophil extracellular traps (NET) network, with the participation of platelets and the process of PMN cell apoptosis. As a result, this affects the lifetime of these cells and the duration of inflammation. We know that various platelet functions may activate an inflammatory response that may result in worsened pathogenesis of many diseases, such as rheumatoid arthritis or cardiovascular diseases. Platelets usually affect neutrophils, and by increasing the production of leukotrienes (LT) such as LTC4, LTD4, and LTE4, this may increase vascular permeability, causing exudate and oedema. The activation state of platelets includes change of their shape from discoid to circular, with numerous pseudopodia [41,52,53] (Figure 1).

Platelet involvement in different medical conditions is described in the context of bacterial and viral infection, sepsis, cardiological conditions, oncologic process, rheumatoid arthritis, and related autoimmune diseases; diabetes and obesity, spleen or kidney damage, neurological diseases of the central nervous system, lung diseases, especially asthma and pneumonia; atherosclerosis and vascular diseases; and also inflammation of the mouth, large intestine, and kidneys. Currently, it is widely recognized that platelets represent a hallmark of immunity and may be considered key players in the inflammatory response as a consequence of their capacity to interact with almost all immune cells. Current understanding of platelet activation pathways and their role as biomarkers could provide new realms of both diagnostic and therapeutic possibilities in evaluation of disease activities and responses to treatment [52,53].

## 5. Platelets in OA

In the past, OA has been described as a local disorder limited to articular cartilage. Recently, OA has been treated as a low-grade inflammatory systemic disease affecting the synovium, synovial fluid, cartilage, subchondral bone, and muscles [54,55,56,57]. Platelets engaged in the inflammatory process, vascular integrity, and hemostasis may play a significant role in OA. Nevertheless, the connection between OA and platelets is still indistinct [54]. Biochemical markers associated with the development of OA are products of cartilage and bone degeneration, such as type II collagen, aggrecan, matrix MMPs, and procollagen type I. Additionally, proinflammatory and anti-inflammatory (IL-6, IL-1B, TNF-a, IL-10, IL-13, and IL-4) molecules are related to the progression of OA by angiogenesis and chemotaxis. Most of them are produced, stimulated, connected, or released by platelets [58,59].

Platelets contain three types of granules: α-granules, dense granules, and lysosomes. α-granules play the main role in inflammation. They interact with leukocytes and vascular cells directly, as well as indirectly, by the secretion of mediators such as cytokines and chemokines. The predominant role of cytokines is in the regulation of the inflammatory response, such as the proliferation and differentiation of lymphocytes [60]. Proinflammatory cytokines secreted by platelets are IL-1, TGF-β, platelet-derived growth factor (PDGF), macrophage inflammatory protein-1a (MIP-1a), and TNF-α [60,61,62]. Chemokines called chemotactic cytokines are signaling proteins released from granules after platelet activation, and have a major role in the activation of the immune response. α-granules secrete many chemokines, such as CXCL1 (β-tromboglobulin), CXCL4 (PF4, platelet factor 4), CCL5 (RANTES, regulated upon activation and normal T cell expressed and secreted), and CXCL12 (SDF-1, stromal cell-derived factor-1) [60,63]. They recruit and differentiate lymphocyte T, and activate neutrophils and phagocytosis of macrophages [60].

Another important molecule produced by α-granules is P-selectin (CD62P), which is also a marker of platelet activation. P-selectin glycoprotein ligand-1 (PSGL-1) connects to P-selectin, and this complex mediates platelet adhesion to leukocytes and monocytes. As an effect, it leads to the recruitment of leukocytes during inflammation [60,64,65]. P-selectin also plays a role in the tethering and rolling of leukocytes [65].

Microparticles (MPs) are platelet-derived vesicles present in arthritic joint fluid. They contain signaling molecules and membrane receptors, alongside activating synovial fibroblasts. MPs induce the production of inflammatory mediators which play an integral role in the development of osteoarthritis [62,63]. They have been recognized as markers of platelet activation, since fragmentation of MPs spontaneously occurs during platelet aggregation and during interaction with monocytes and leucocytes [64,66]. MPs, by releasing IL-1α and IL-1β, stimulate fibroblast-like synoviocytes (FLS) to secrete IL-6 and IL-8 [52,62,67]. MPs are released by the signaling receptor glycoprotein VI (GPVI), which triggers MP production by stimulation of IL-1. GPVI is the predominant collagen receptor expressed by megakaryocytes and platelets [66]. Platelets are activated by collagens I, III, IV, V, VI, and XIII. Among these, collagens I, III, and IV are more reactive with platelets, and induce strong adhesion [68]. The soft tissue surrounding the joint consists mostly of collagens I and III, and it is known that platelets react with them deep below the subendothelium [63]. Platelets and chondrocytes generate reactive oxygen species (ROS). The basal level of intercellular ROS is enlarged by collagen GPVI stimulation. An increased amount of ROS leads to the activation of platelets and thrombus formation, and then to cartilage destruction as well as inhibition of cartilage synthesis [54,69].

Platelet number is associated with plasma IL-1β concentration. IL-1β stimulates platelets to form an autocrine stimulatory loop by platelets expressing the IL1R1 receptor [67]. IL-1β is a proinflammatory cytokine and a suppressor of type II collagen and aggrecan synthesis [58]. In addition, IL-1β increases the expression of MMPs and induces production of a cartilage catabolic mediator, nitric oxide [70,71], which inhibits platelet aggregation [72]. Moreover, some studies showed mild to moderate correlation of IL-1β concentration and joint space width or joint alignment [73], in addition to increased risk of familial OA [71]. A negative correlation between a decrease in pain severity and a change of IL-1β in synovial fluid was observed [74]. IL-1β also stimulates the production of IL-6 and IL-8, which are also proinflammatory cytokines entailed in platelet hyperactivation [70,75].

As a multifunctional cytokine, IL-6 regulates the acute phase of the inflammatory and immune responses. Biological activities of IL-6 are promoted by trans-signaling, which results in conversion from acute to chronic inflammation [75,76]. Signal transduction is mediated by two molecules: membrane-bound β-receptor glycoprotein 130 present on platelets (gp130), and IL-6 receptor (IL-6R) [75]. IL-6 builds a complex with soluble forms of IL-6R; then, this complex binds to gp130 [76]. Gp130 is expressed, inter alia, on platelets. The presence of IL-6 leads to platelet-derived IL-6 trans-signaling [75], which causes platelet hyperactivation [77]. Higher platelet activation leads to survival and leukocyte activation [57]. IL-6 is also a mediator of platelet hyperactivity, and by acting directly on megakaryocytes, increases platelets in circulation and moreover initiates the coagulation cascade [54,77,78]. In animal models, IL-6 decreased the production of collagen type II [79]. Livshits et al. demonstrated a relationship between IL-6 and the increased prevalence of OA diagnoses in long term studies (15-year follow-up) [80]. Stannus et al. found associations between the circulating levels of IL-6 and TNF-α and cartilage loss in knee OA in addition to joint space narrowing in the 3-year follow-up [59].

Mean platelet volume (MPV) is suggested to be a marker of low-grade inflammation in OA [81]. MPV reflects platelet size and platelet activity, and additionally regulates thrombopoesis in conjunction with other inflammatory cytokines (IL-1, Il-6, and TNF-a) which are found in OA serum [57,81]. Balbaloglu et al. [81] investigated MPV levels in reference to patients with synovitis in knee OA. They compared MPV results in three groups—patients with synovitis associated to knee osteoarthritis, patients with knee osteoarthritis, and a control group (patients who did not have joint complaints). MPV blood levels significantly differed between the patients with synovitis associated with osteoarthritis and the patients with knee osteoarthritis; the levels also differed with the control group. However, there was no difference between OA and control. This suggests that MPV may be useful as a marker for inflammation in osteoarthritis in synovitis patients [81]. Another study analyzed MPV as a predictor of hip OA. Retrospectively, radiographic images were classified according to the Kellgren–Lawrence (KL) grading score. In statistical analyses, MPV was significantly higher in KL grades 3–4 compared with KL 1–2 [57]. Kwon et al. [54] investigated the correlation between platelet count and knee or hip osteoarthritis in Korean women over 50 years of age. The KL grading system was used for evaluating the OA grade. This study showed significantly higher platelet and WBC counts in patients with symptomatic OA, confirmed by radiography. The researchers also found a linear relationship between OA and platelet count [54].

TNF-α, a proinflammatory cytokine, activates platelets by interaction with tumor necrosis factor receptor superfamily member 1A (TNFRSF1A), and additionally stimulates the arachidonic acid pathway. Other cells witch release TNF-α may exert agonist activity for platelets [59]. Breda et al. found that a higher level of TNF receptors was associated with increased pain, stiffness, and moreover worse radiographic scores in osteoarthritis [82].

Extended platelet interaction with cells involved in the inflammatory process may lead to an adverse effect as a result of overstimulation of the immunological system [60]. Prolonged inflammation leads to destruction of cartilage, bone, and ligaments.

## 6. Platelets as a Treatment for OA

Platelet-rich plasma as a minimally invasive therapy for OA is currently well known and widely used. It is a safe, low-cost, and effective means as an alternative non-surgical treatment for knee osteoarthritis. PRP consists of autologous plasma with platelets, which induce secretion of growth factors such as PDGF, TGF-β1, TGF-β2, insulin-like growth factor (IGF), epidermal growth factor (EGF), VEGF, and FGF. Platelets also contain cytokines and immunomodulators. As a result of their anti-inflammatory nature and potential role in tissue healing, they are a research target for the potential treatment for osteoarthritis [83,84,85]. Several authors demonstrated that platelets promote healing and modulate inflammation, both in vitro and in vivo [86,87,88]. TGF-β is important for chondrogenesis by inducing cell proliferation in cartilage, osteochondrogenic differentiation, and by increasing gene expression of aggrecan and collagen II [89]. PDGF stimulates cell proliferation and production of proteoglycan [90], and FGF induces chondrocyte and MSC proliferation [91]. Several GFs have a synergistic effect in the OA joint.

PRP is activated by the addition of calcium chloride or natural collagen from damaged tissue, which becomes prepared to secrete granules and form a clot. Platelet-rich fibrin (PRF) does not require activators, and the clotting process occurs naturally. In addition, both PRP and PRF may contain leukocytes. This process yields four main products (pure PRP, leukocyte-rich PRP, pure PRF, and leukocyte-rich PRF), each of which has a different mechanism of action, contains different molecules, and should be used in different indications [84,92].

An in vitro study performed by Osterman et al. created a coculture system with synovia and cartilage harvested from patients during total knee arthroplasties. Two preparations were used—leukocyte-poor and leukocyte-rich PRP. Both preparations decreased the expression of genes correlated with the inflammatory process in OA (TIMP-1, ADAMTS-5) cartilage and synovia, and increased aggrecan expression compared to non-OA cartilage. Moreover, nitric oxide production, which is an indicator of inflammation, was significantly lower in both PRP groups [87].

A similar study compared cartilage and synovia from patients undergoing TKA and cocultured with HA or PRP. An increaseaggrecan gene expression and decreased COL1A1 in the PRP group were observed in comparison to HA. A decreased volume of TNF-α in both PRP and HA was observed in comparison to the control, and IL-6 concentration was significantly decreased in the HA group. IL-1B was decreased in all treatment groups. There was no difference in cartilage MMP-13 expression in all samples of cartilage, but MMP-13 expression in synoviocytes decreased in the PRP group compared to the control and HA groups. Likewise, an increase in synoviocyte gene expression of HAS-2 was observed in the PRP group, but there was no difference in TNF-α in all three groups [93].

Lacko et al. investigated the effect of three PRP injections in patients with unilateral knee OA. This study showed an improvement in the WOMAC score and a reduction in the VAS score for pain at the 3-month follow-up. The most important finding was a significant decrease in proinflammatory markers (eotaxin, MCP-1, MMP-1, IL-1o, EGF, PDGF, and TGF-B) compared to baseline level, and an increase in pro-anabolic markers (BMP-2, COMP, collagen type II, and GRO) of the cartilage and anti-inflammatory markers [94].

Tang et al. compared PRP injection versus hyaluronic acid (HA) for patients with OA. In the 20 randomized controlled trials (RCTs) enrolled in their meta-analysis in 2020, it was shown that PRP has a better analgesic effect than HA in long-term recovery (at the 6-month and 12-month follow-ups) and also leads to better functional recovery in 12 months. The risk of adverse events was not increased compared with HA [95]. Similar observations were noted by Trams et al. in their meta-analysis [96].

Kon et al. performed a systematic review of PRP injections in OA patients. Twelve meta-analyses were included, comparing PRP injections to HA, corticosteroids (CS), and placebo. The main finding showed an advantage of PRP over other injections in the 12-month follow-up. The main limitation of the study was in the variability of the applied PRP, including harvest blood volume, preparation method, and number of injections [85].

## 7. Platelets in OA—An Updated Meta-Analysis

In 2020 we published an article, “The Clinical Use of Platelet-Rich Plasma in Knee Disorders and Surgery—A Systematic Review and Meta-Analysis” [96]. In this article we have updated the chapter titled “osteoarthritis”.

A literature search of electronic databases identified a total of 386 records according to the selected search algorithm. There were 355 citations excluded as irrelevant, according to title and/or abstract. The abstracts of 31 remaining articles were assessed for eligibility. From these, 21 were excluded. The remaining 10 clinical studies published between 2020 and 2022, with a total of 974 patients, were included in this review. The literature search flowchart is shown in Figure 2.

Forty-eight studies (published between 2005 and 2022), including 3936 patients, evaluated the use of PRP in osteoarthritis treatment. Follow-up ranged from 6 months up to 2 years.

Thirty-seven studies compared PRP versus control groups [88,97,98,99,100,101,102,103,104,105,106,107,108,109,110,111,112,113,114,115,116,117,118,119,120,121,122,123,124,125,126,127,128,129,130,131,132], six studies compared PRP with the addition of another substance (MSC or HA) versus control groups [122,123,133,134,135], six studies compared multiple injections of PRP [100,118,119,136,137,138], five studies compared PRGF-Endoret versus control groups [125,128,139,140,141], two studies compared autologous conditioned plasma (ACP) versus control groups [142,143], and one study compared intraosseous injection versus intra-articular injection versus the control group [108]. In twenty-nine studies, HA [101,103,104,106,107,108,109,110,112,113,114,115,116,117,118,120,122,123,125,126,127,128,132,136,139,141,143] was used as a control; in eleven studies placebo was used as a control (saline, no injection, physical therapy) [98,99,111,118,120,122,124,133,136,137,138]; in five studies corticosteroids [88,100,113,129,134] were used as the control; and in two studies acetaminophen [97,105] was used as the control. One study used peptides [132], ozone [128], dextrose prolotherapy [130], or bone marrow aspirate concentrate [127].

Thirty-two studies reported pain via the VAS comparing PRP versus placebo [97,98,99,117,123,130,135], dextrose [129] corticosteroids [88,100,128], or HA [101,102,103,104,106,107,108,109,110,113,116,118,122,124,125,126,127,131,132,138,143] (Figure 3). Placebo and HA subgroups showed significant differences in favour of PRP (*p* = 0.02; *p* < 0.00001); moreover, the steroid group showed significant differences in favour of PRP (*p* = 0.04), despite the dextrose subgroup showing significant differences in favour of the control group (*p* < 0.00001). The pooled estimates for these studies also showed significant differences in favour of PRP (*p* = 0.002).

Functional outcome was measured in thirty-five studies via the WOMAC scale. One study was excluded from the meta-analysis as a consequence of only reporting WOMAC pain scores [104]. Thirty-four studies compared PRP versus control groups: placebo [97,98,99,105,111,117,119,120,123,130,141], corticosteroids [100], dextrose [129], or HA [103,106,108,109,110,113,115,116,118,121,122,124,125,126,127,131,132,138,139,140,142] (Figure 4). The pooled estimates for these studies showed significant differences in favour of PRP (*p* < 0.00001); furthermore, each subgroup showed significant differences in favour of PRP (*p* < 0.00001).

Ten studies evaluated functional outcomes in IKDC rating scores, and showed significant differences in favour of PRP (*p* < 0.0001) (Figure 5). Seven studies showed significant differences in favour of PRP compared to HA as a control group (*p* = 0.0004) [101,104,107,115,125,126,143]; one study compared PRP with corticosteroids and [128] showed significant differences in favour of the experimental group (*p* < 0.0001); and two studies showed non-significant differences in favour of PRP when compared with the placebo (*p* = 0.24) [120,135].

Eight studies evaluated osteoarthritis outcomes via KOOS scores [88,101,102,114,126,130,133,134] (Figure 6). We excluded from the meta-analysis two of these studies due to a lack of measurements in one [98], and division of the results according to the physician in another [112]. The pooled estimates for these studies showed non-significant differences in KOOS for sport (*p* = 0.93), quality of life (*p* = 0.76), and ADL (*p* = 0.91); KOOS symptoms (*p* = 0.21) and pain (0.91) sub-scales were in favour of the control group (*p* > 0.05).

Functional outcomes were also measured with KSS scores in two studies [109,129]; Lysholm scores in three studies [104,109,123]; Tegner scores in four studies [101,109,133,143]; Outcome Measures in Arthritis Clinical Trials–Osteoarthritis Research Society International (OMERACT–OARSI) pain measure in four studies [98,131,140,141]; and Lequesne scores in nine studies [100,109,123,125,128,132,139,140,141]. Quality of life was measured with SF-36 scores in five studies [98,99,105,115], SF-12 in two studies [97,112], and European Quality of Life (EQoL) in two studies [102,112].

Thirty-two studies reported adverse events (Figure 7). Nine studies [98,99,111,118,120,121,124,131,142] comparing PRP versus placebo reported significant differences in favour of the control groups (*p* = 0.03); twenty studies [101,102,106,108,109,110,112,113,117,119,122,123,125,126,127,133,139,140,141,143] comparing PRP versus HA also reported significant differences in favour of the control groups (*p* = 0.003); furthermore, two studies [88,129] comparing PRP versus steroids (*p* = 0.002), and one study comparing PRP versus MSC [135] showed non-significant differences in favour of PRP (*p* = 0.60). The pooled estimates for these studies showed significant differences in favour of the control groups (*p* = 0.03).

Two studies were at high risk of bias for three domains [126,142], seven studies were at high risk of bias for two domains [105,113,116,118,129,130,137], and sixteen studies were at high risk of bias for one domain [100,103,107,108,109,110,112,114,117,121,123,124,128,131,135,138]. One study was at high risk of performance bias for three domains, with the risk of reporting bias for two domains [115], and seven studies had a lack of possibilities to define bias [99,105,113,115,119,123,127] (Figure 8).

## 8. Materials and Methods

### 8.1. Search Strategy

In this review we concentrated on PRP applications to knee osteoarthritis compared with placebo or other treatment control groups. This study was completed in compatibility with the 2009 Preferred Reporting Items for Systematic Reviews and Meta-Analyses (PRISMA) statement. A systematic review of the use of platelet-rich plasma in knee lesions was completed with a comprehensive published literature search in PubMed, Embase, Cochrane Database of Systematic Reviews, and Clinicaltrials.gov. The references of the investigations found in this search were cross-referenced to identify additional pertinent studies not identified in the original searches. All searches were performed in January 2022. The searches were performed combining the following keywords: (1) “PRP”, or “platelet-rich plasma”, or “plasma rich in growth factors”, or “platelet derived growth factor”, or “platelet derived”, or “platelet gel”, or “platelet concentrate”, or “PRF”, or “platelet rich fibrin”, or “ACP”, or ”autologous conditioned plasma”, or ”PRGF”, or “platelet lysate”; and (2) “knee osteoarthritis” or “osteoarthritis”. This was an update on a systematic review registered with PROSPERO (International Prospective Register of Systematic Reviews, ID 167715).

### 8.2. Inclusion and Exclusion Criteria

This review included all clinical studies meeting the following inclusion criteria: PRP utilization as conservative treatment in knee osteoarthritis, English language, human subjects, paper published in a peer-reviewed journal, and full text available. Only randomized controlled trials were included. Exclusion criteria included all animal studies, basic scientific investigations, case reports, review articles, expert opinions, letters to editor, studies without control groups, studies not using PRP, papers not peer reviewed, papers not in English, trials evaluating platelet-poor plasma, and investigations on other diseases unrelated to the knee joint. The investigations included in this study were independently reviewed by two orthopedic surgeons/authors for inclusion and exclusion criteria.

### 8.3. Types of Interventions

We compared intralesional, injected PRP preparation with the following:-Placebo injection (low-volume saline injection, matching the PRP volume);-High-volume saline image-guided injection;-Local steroidal injection;-Hyaluronic acid injection;-Exercise and other physical therapies (e.g., low-dose radiation therapy, eccentric loading program, dry needling);-Any other medications administered locally or systemically aimed at treating pain; and-Combinations of the active interventions listed above.

### 8.4. Outcomes

Primary outcomes included the following:-Pain as measured by standard validated pain scale, such as visual analogue score (VAS), EQ-VAS, or numerical rating scale (NRS);-Functional measurement by any standard validated scale, such as the International Knee Documentation Committee (IKDC), Western Ontario and McMaster Universities Osteoarthritis Index (WOMAC), Knee Society Score (KSS), Victorian Institute of Sport Assessment (VISA), 36-Item Short Form Survey (SF-36), Knee injury and Osteoarthritis Outcome Score (KOOS), Lysholm Knee Scoring Scale, Tegner Activity Score, and Ikeuchi grade knee rating scale.

### 8.5. Data Collection and Analysis

For each study included in the analysis, the following data were extracted by two independent reviewers: authors, year of publication, type of knee lesions, details of interventions in the study, sample size (randomized and analysed), outcome measurements, follow-up period, main results, and percentage and type of adverse events included in the publication. Each study’s level of evidence was examined and evaluated based on criteria established by the Oxford Centre for Evidence-Based Medicine Levels of Evidence Working Group [109]. Measures of treatment effect at the final point used were the means and standard deviations for continuous outcome measures. When studies reported other measures (e.g., median) and other dispersion measures, such as standard error (SE) of the mean or 95% CI of the mean, range or interquartile range (IQR), we calculated the SD in order to perform the relevant meta-analytical pooling according to previous studies [144,145].

The study weight was calculated using the Mantel–Haenszel method. We assessed statistical heterogeneity using Tau2 or Chi2, df, and I2 statistics. The I2 statistic describes the percentage of total variation across trials that is a result of heterogeneity. In the case of low heterogeneity (I2 < 40%), studies were pooled using a fixed-effects model; otherwise, a random-effects analysis was performed.

Subgroup analysis was undertaken for the type of clinical trial and for the type of control intervention.

### 8.6. Risk of Bias Assessment

Revised Cochrane risk-of-bias tool was used to evaluate risk. Disagreement in the risk of bias assessment was resolved by consensus and, if necessary, by the opinion of a third reviewer. A study was deemed to be:-“Low risk” if all items were scored as “low risk”;-“Moderate risk” if up to two items were classified as “high risk” or “unclear risk”;-“High risk” if more than two items were scored as “high risk”.

We presented our assessment of risk of bias using two “Risk of Bias” summary figures for every subsection of the manuscript.

### 8.7. Statistical Analysis

Qualitative statistical analysis and meta-analysis were performed using R software and REVMAN 5.4 [145,146], with *p*-values of <0.05.

## 9. Conclusions

OA is a degenerative joint disease primarily associated with aging. Since affected patients are mainly characterized by cartilage malfunction, inflammation plays an essential role in the pathogenesis of OA. Although there are many well-described molecular mechanisms and mediators involved in inflammation, it still remains controversial whether inflammatory mediators are primary or secondary regulators of the cartilage damage and disturbed repair processes in OA. The latest knowledge about the activation of inflammatory mediators could play a significant role in the early diagnosis of inflammation in OA and become potentially useful in therapeutic protocols.

Given the pivotal role of platelets in hemostasis, inflammation, and the immune response, it can be assumed that platelets also contribute to inflammation in OA; however, their exact role in this process requires further investigation.

Joint diffusion as a result of inflammation is commonly observed in other inflammatory arthropathies, including RA; in contrast to OA, the role of platelets in RA has been established. In RA, the platelet count is related to disease activity, and thrombocytosis is commonly detected in the active phase of the disease.

Of special interest are the multiple mechanisms involved in platelet activation in osteoarthritis, which offer numerous options for potential arthritis therapies.

As stated in this review, the role of platelet-rich plasma is a well described, commonly used, minimally invasive method for knee osteoarthritis treatment. Multiple randomized controlled trials described PRP as a safe, easily accessible, and low-cost procedure as an alternative to other non-surgical treatments for osteoarthritis used as various control groups. There is still a need for a standardized PRP injection to be established from the multitude of products available for medical usage. Moreover, there is still a lack of studies for using PRP as an alternative to surgical treatment, since the majority of trials only compare PRP to other intra-articular injections. Platelets may have a potential healing role as a result of their proinflammatory and anti-inflammatory functions.

## Figures and Tables

**Figure 1 cells-11-01080-f001:**
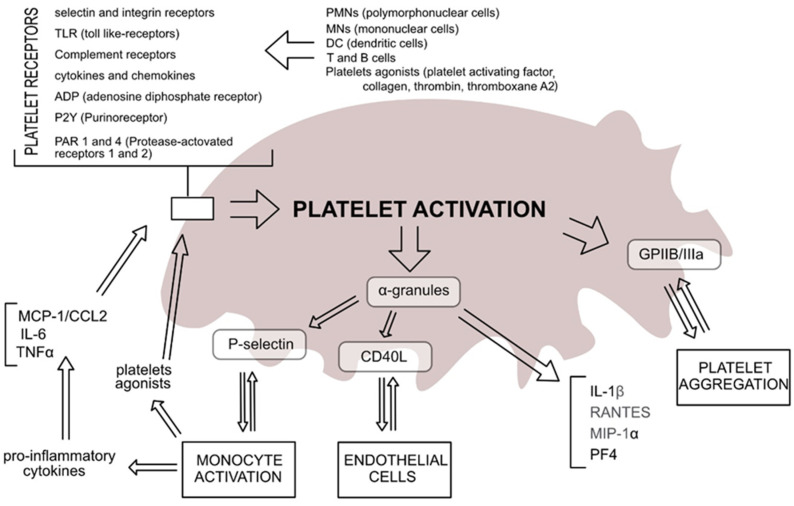
Molecular pathways involved in platelet activation during the inflammatory process.

**Figure 2 cells-11-01080-f002:**
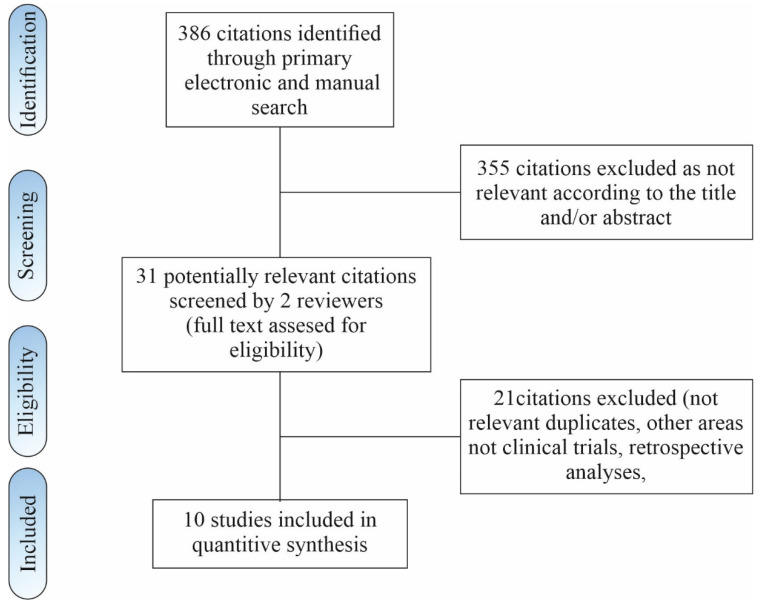
Flow chart of study inclusion.

**Figure 3 cells-11-01080-f003:**
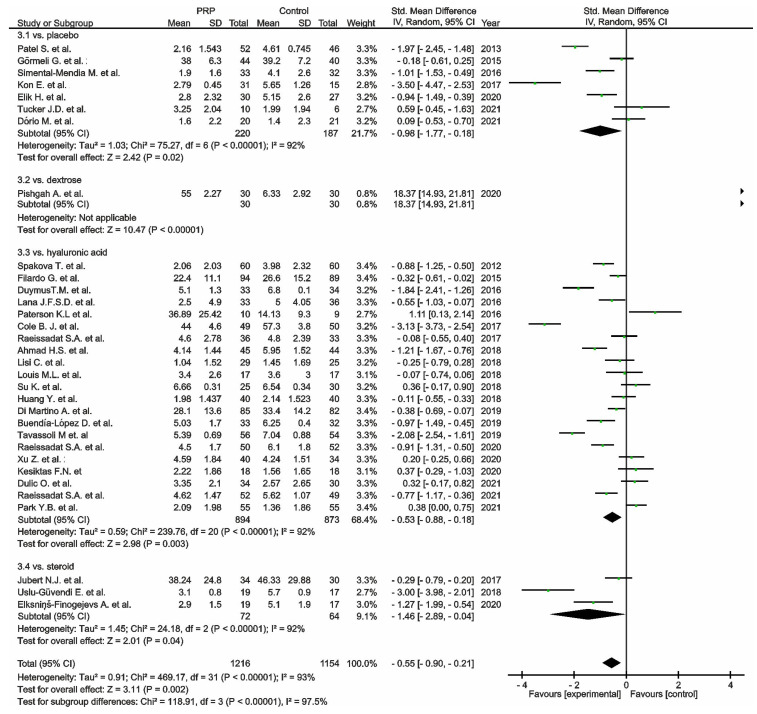
Forest plot for VAS scores comparing PRP versus control (CI, confidence interval; IV, inverse variance; SD, standard deviation; 
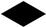
, mean with 95% confidence interval for total and subtotals; 
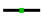
, mean with 95% confidence interval for source data).

**Figure 4 cells-11-01080-f004:**
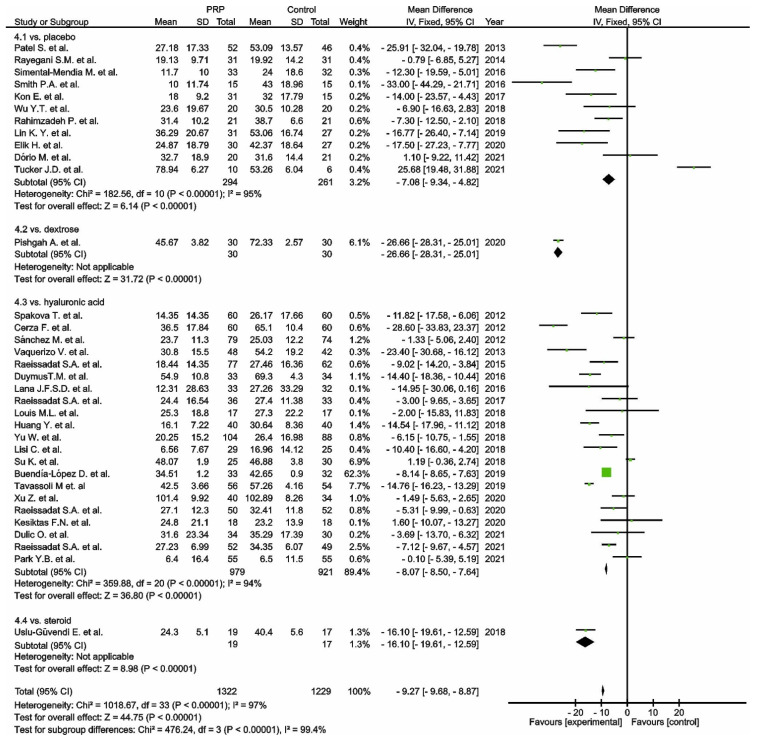
Forest plot for WOMAC scores (CI, confidence interval; IV, inverse variance; SD, standard deviation; 
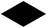
, mean with 95% confidence interval for total and subtotals; 
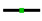
, mean with 95% confidence interval for source data).

**Figure 5 cells-11-01080-f005:**
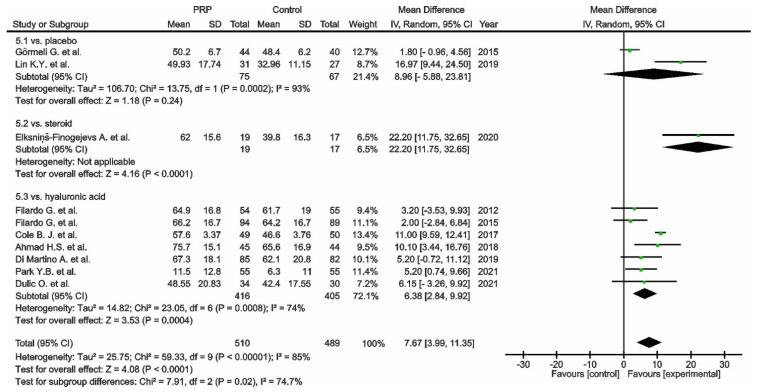
Forest plot for IKDC scores (CI, confidence interval; IV, inverse variance; SD, standard deviation; 
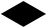
, mean with 95% confidence interval for total and subtotals; 
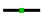
, mean with 95% confidence interval for source data).

**Figure 6 cells-11-01080-f006:**
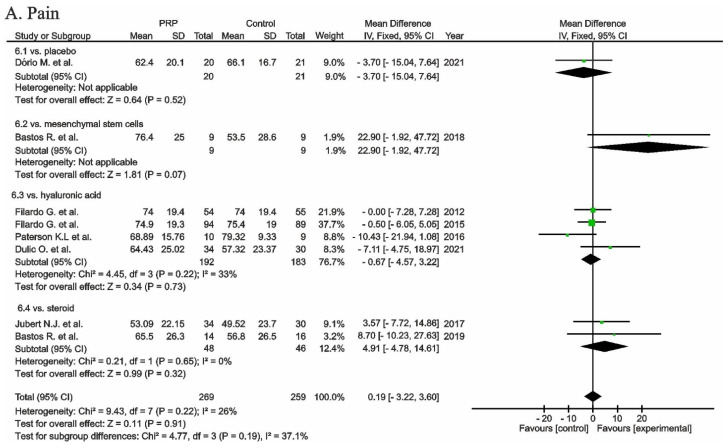
Forest plot for KOOS sub-scores (ADL, activities of daily living; CI, confidence interval; IV, inverse variance; QoL, quality of life; SD, standard deviation; 
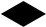
, mean with 95% confidence interval for total and subtotals; 
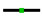
, mean with 95% confidence interval for source data).

**Figure 7 cells-11-01080-f007:**
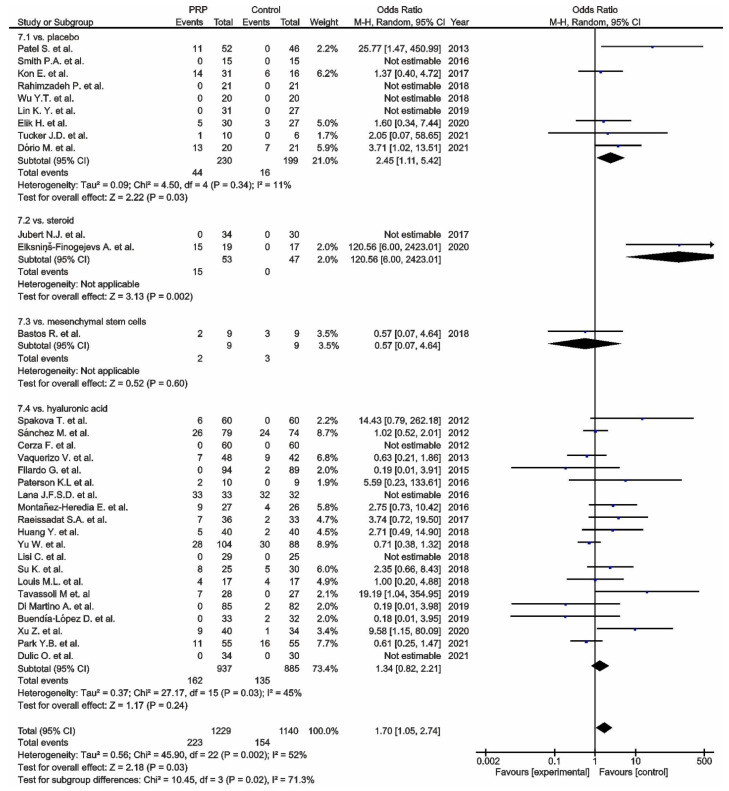
Forest plot for adverse events (CI, confidence interval; M-H, Mantel-Heanszel; 
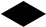
, odds ratio with 95% confidence interval for total and subtotals; 
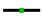
, odds ratio with 95% confidence interval for source data).

**Figure 8 cells-11-01080-f008:**
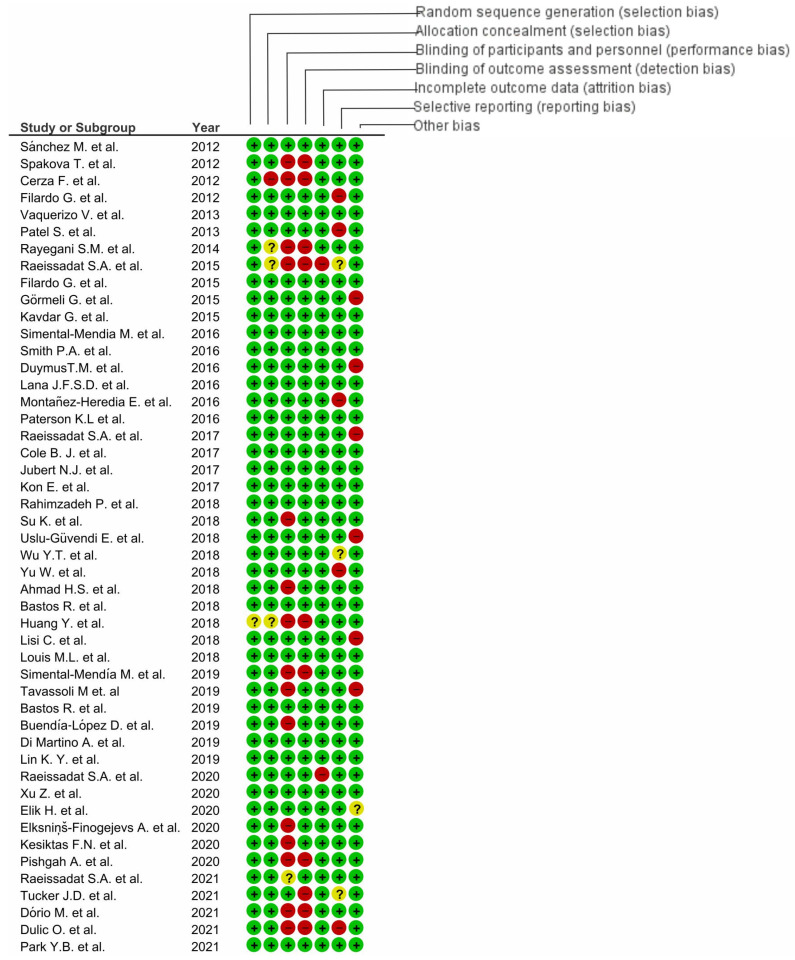
Risk of bias analysis for PRP application in osteoarthritis (
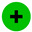
, low risk of bias; 
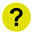
, unclear risk of bias; 
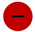
, high risk of bias).

## Data Availability

Not applicable.

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
