# Peer review of "Role of Platelets in Osteoarthritis—Updated Systematic Review and Meta-Analysis on the Role of Platelet-Rich Plasma in Osteoarthritis"

_cells, 2022, doi:10.3390/cells11071080_

Round 1
Reviewer 1 Report
This review aims to discuss a role of platelets in arthritis. The authors describe basic functions of platelets and connect those to osteoarthritis (OA) and rheumatoid arthritis (RA). They describe also treatment options for platelets in OA. In addition, this review contains an updated meta-analysis on the role of platelet rich plasma (PRP) in OA.
This review is basically assembled of two independent parts: a review of original data on platelets in OA and RA and an updated meta-analysis on PRP in the context of OA. The meta-analysis – as implied by the term “updated” has been published in parts already by the same group (Trams et al., Life 2020, doi: 10.3390/life10060094) diminishing originality. In its present form, the review is lengthy, unfocussed and poorly structured.
Major points
-The first part of the manuscript is too extended and needs to be significantly shortened. Specifically:
- 1. Overview: This part can be shorten profoundly and it is sufficient to refer to some recently published real good reviews on OA.
- As the main topic is inflammation and platelets, the review should start with chapter 2. Inflammation-role in pathogenesis in OA.
- Chapter 5. Platelets in OA: The last paragraph of this chapter on page 7 is out of scope as it is not connected to the topic “platelets” and should thus be removed.
- As the second part- the meta-analysis is strictly on OA- all parts referring to RA should be removed. Including RA data makes the review unfocused and incomplete. This refers to chapter 6. “Platelets in RA” which should be completely removed and instead it should be concentrated on OA solely.
- Chapter 6 (presumably a mistake and should have been chapter 7) “Platelets as treatment of OA” needs to be restructured and significantly shortened. Right now it is confusing as it mixes up different types of studies. It should be organized according to in vitro studies, preclinical and clinical studies in that order. In addition, all text, which refers to RA needs to be removed for the sake of focus.
Specific points
- 3 appears wrong in this context.
- 20 on page 3 is too old and does not fit in the context.
- Reference is missing for paragraph 1 of chapter 3.
- Reference is missing for paragraph 3 on page 4.
- Sentence 1, paragraph 2 on page 5: It must be “collagen fragments”.
- Chapter 5: It must be “OA” and not “AO”.
- Last sentence on page 6 and first sentence on page 7: Please define the “control group”. In addition, which joint(s) is meant for the group “patients with synovitis associated with OA”? Because this is not a discriminating criterion from the group “patients with knee OA” as they also can have synovitis!
- Figure 6 is unreadable!
Author Response
First of all, thank you for your comments and suggestions that allowed us to greatly improve the quality of the article. We agree with all your comments, and we modified the manuscript point by point accordingly.
- The first part of the manuscript is too extended and needs to be significantly shortened. Specifically:
- 1. Overview: This part can be shorten profoundly and it is sufficient to refer to some recently published real good reviews on OA.
- We modified and shorten Overview section: lines 28-91 were removed, chapter Osteoarthritis was rewritten, new references were added
- As the main topic is inflammation and platelets, the review should start with chapter 2. Inflammation-role in pathogenesis in OA.
- We modified the manuscript and we started the review with a chapter: Inflammation-role in pathogenesis in OA.
- Chapter 5. Platelets in OA: The last paragraph of this chapter on page 7 is out of scope as it is not connected to the topic “platelets” and should thus be removed.
- the lines 400-425 were removed
- As the second part- the meta-analysis is strictly on OA- all parts referring to RA should be removed. Including RA data makes the review unfocused and incomplete. This refers to chapter 6. “Platelets in RA” which should be completely removed and instead it should be concentrated on OA solely.
- All parts of manuscript referring to RA were removed (lines 115-122 and 426-489)
- Chapter 6 (presumably a mistake and should have been chapter 7) “Platelets as treatment of OA” needs to be restructured and significantly shortened. Right now it is confusing as it mixes up different types of studies. It should be organized according to in vitro studies, preclinical and clinical studies in that order. In addition, all text, which refers to RA needs to be removed for the sake of focus.
- We reconstructed and shortened Chapter 6 according to Your suggestions. This chapter is more clearly organized. (lines 512-535)
Minor:
- 3 appears wrong in this context. - reference was updated (chapter Osteoarthritis was rewritten, new references were added – lines 130-160)
- 20 on page 3 is too old and does not fit in the context. (this part was removed – lines 126-129)
- Reference is missing for paragraph 1 of chapter 3. (We added references no 5, 21-23; line 169)
- Reference is missing for paragraph 3 on page 4. (We added reference no 18,21,28 and 29-32 , line 201 and 206)
- Sentence 1, paragraph 2 on page 5: It must be “collagen fragments”. - corrected
- Chapter 5: It must be “OA” and not “AO”. - corrected
- Last sentence on page 6 and first sentence on page 7: Please define the “control group”. In addition, which joint(s) is meant for the group “patients with synovitis associated with OA”? Because this is not a discriminating criterion from the group “patients with knee OA” as they also can have synovitis!
- We clarify last sentence on page 6: control group is defined as patients who did not have joint complaints, as well as we define joint (knee) for patients with synovitis associated with OA according to the original article. (lines 374-375)
- Figure 6 is unreadable! - New figure 6 was created
Another round of proof reading was conducted by native speaker editing services.

Reviewer 2 Report
Major point
The manuscript seems to be a combination of manuscripts shared by several authors. The same terms and abbreviations appear over and over again. The person in charge should oversee and organize the entire manuscript. The content of the latter part seems to be good. Unfortunately, there seems to be a confusion in the understanding of the regulation of platelets in pathological inflammatory states.
On page 7, there is a description of IL-1β and IL-6, but there is some confusion here. It is IL-6 that regulates platelet counts in chronic inflammatory conditions such as rheumatoid arthritis. Platelet count decreases when anti-IL-6 receptor antibodies are administered to patients with rheumatoid arthritis.(Incidentally, the platelet regulator in healthy individuals is not IL-6.)
On page 7, the authors state,"IL-1β stimulates chondrocytes to increase production of IL-6, by binding two molecules: membrane-bound β-receptor glycoprotein 130 present on platelets (gp130) and IL-6R (IL-6 receptor)" which is incorrect. Molecules that have been found to share gp130 as a signal transducer are IL-6, IL-11, IL-27, IL-35, IL-39, LIF, OSM, CT-1, CNTF, and CLCF1. (Not IL-1β)
With the exception of some cells, such as hepatocytes, the IL-6 receptor is not located on the cell membrane. The soluble form IL-6 receptor binds to IL-6 and the complex can transduce the signal to gp130.
Although this manuscript is on the topic of platelets and arthritis, it does not adequately and incorrectly describe the consequences and mechanisms of high levels of IL-6, a pathological platelet hyperplasia factor, in patients with arthritis.
Minor points
1. Please,review where to show abbreviations. In many journals, each abbreviation shows where it first appear in the text (not including the abstract). This manuscript is confusing because some abbreviations are indicated in the abstract and some are shown in the text. If MIP is explained in the text, the same manner should be done for "IL" and "CXCL". (Check page 3, line 37, and page 8, line 12.)" The terms "interleukin-1 (IL-1)," "transforming frowth factor (TGF)," and "macrophage inflammatory protein (MIP)" appear several times.)
2. Please, change "Interleukins (tumor necrosis factor α (TNFα), IL-6, ..." to "cytokines (tumor necrosis factor α [TNFα], interleukin [IL-] 6," on line 6 of the abstract.
3. In line 12 of the abstract, the words "review and meta-analyse" are incorrect. Since "analyse" is a verb, the plural noun "analyses" is correct here.
4. On page 6, line 2, I think "onocologic" is an error for "oncologic".
5. On page 6, line 13, what is AO ?
6. On page 6, line 26, please correct "Il-6" to IL-6.
7. On page 6, line 14, the sentence "Recently AO has been described as....." is too eccentric. It would be better to change the wording.
7. On page 7, line 20, "IL-1βin" and "IL-1βstimulates" should be changed to "IL-1β in" and "IL-1β stimulates".
8. Page 7, "GP130" should be changed to "gp130".
Author Response
First of all, thank you for your comments and suggestions which allowed us to greatly improve the quality of the article. We agree with all your comments, and we modified the manuscript point by point accordingly.
The manuscript seems to be a combination of manuscripts shared by several authors. The same terms and abbreviations appear over and over again. The person in charge should oversee and organize the entire manuscript. The content of the latter part seems to be good. Unfortunately, there seems to be a confusion in the understanding of the regulation of platelets in pathological inflammatory states. – Article was overseen and modified accordingly to your suggestions
On page 7, there is a description of IL-1β and IL-6, but there is some confusion here. It is IL-6 that regulates platelet counts in chronic inflammatory conditions such as rheumatoid arthritis. Platelet count decreases when anti-IL-6 receptor antibodies are administered to patients with rheumatoid arthritis.(Incidentally, the platelet regulator in healthy individuals is not IL-6.)
On page 7, the authors state,"IL-1β stimulates chondrocytes to increase production of IL-6, by binding two molecules: membrane-bound β-receptor glycoprotein 130 present on platelets (gp130) and IL-6R (IL-6 receptor)" which is incorrect. Molecules that have been found to share gp130 as a signal transducer are IL-6, IL-11, IL-27, IL-35, IL-39, LIF, OSM, CT-1, CNTF, and CLCF1. (NotIL-1β). With the exception of some cells, such as hepatocytes, the IL-6 receptor is not located on the cell membrane. The soluble form IL-6 receptor binds to IL-6 and the complex can transduce the signal to gp130. Although this manuscript is on the topic of platelets and arthritis, it does not adequately and incorrectly describe the consequences and mechanisms of high levels of IL-6, a pathological platelet hyperplasia factor, in patients with arthritis. - We rewrite paragraph with action of IL-6 on platelets. (lines 306-368)
Minor points
- Please,review where to show abbreviations. In many journals, each abbreviation shows where it first appear in the text (not including the abstract). This manuscript is confusing because some abbreviations are indicated in the abstract and some are shown in the text. If MIP is explained in the text, the same manner should be done for "IL" and "CXCL". (Check page 3, line 37, and page 8, line 12.)" The terms "interleukin-1 (IL-1)," "transforming frowth factor (TGF)," and "macrophage inflammatory protein (MIP)" appear several times.) - We have also reviewed the abbreviations and some abbreviations have been replaced by more commonly used ones
2. Please, change "Interleukins (tumor necrosis factor α (TNFα), IL-6, ..." to "cytokines (tumor necrosis factor α [TNFα], interleukin [IL-] 6," on line 6 of the abstract. - corrected
3. In line 12 of the abstract, the words "review and meta-analyse" are incorrect. Since "analyse" is a verb, the plural noun "analyses" is correct here. - corrected
4. On page 6, line 2, I think "onocologic" is an error for "oncologic". - corrected
5. On page 6, line 13, what is AO ? – corrected , it should be “OA”
6. On page 6, line 26, please correct "Il-6" to IL-6. - corrected
7. On page 6, line 14, the sentence "Recently AO has been described as....." is too eccentric. It would be better to change the wording. - rewritten
7. On page 7, line 20, "IL-1βin" and "IL-1βstimulates" should be changed to "IL-1β in" and "IL-1β stimulates". - corrected
8. Page 7, "GP130" should be changed to "gp130". - modified

Round 2
Reviewer 1 Report
The authors have done a good job and have addressed all my comments properly. The manuscript really improved!
Some minor comments remain to be addressed.
Abstract
- As the review is now focused on osteoarthritis, the word “arthritis” should be replaced by “osteoarthritis”. This should be handled consistently throughout the manuscript.
Main text body
- I would rearrange chapter 2 and 1. Chapter 2 “Osteoarthritis” needs to be the first chapter as it introduces the disease in general. “Inflammation- role of pathogenesis of OA” should be then chapter 2. I guess it was a misinterpreting of my comments to place the introducing chapter about "osteoarthritis" as chapter 2.
- To this end, there is a misnumbering of chapters. “Materials and methods” should be chapter 8 and not chapter 9! Consequently, chapter 10 –conclusions- needs to be then chapter 9.
Author Response
First of all, thank you for your comments and suggestions which allowed us to greatly improve the quality of the article. We agree with all your comments, and we modified the manuscript point by point accordingly.
Abstract
- As the review is now focused on osteoarthritis, the word “arthritis” should be replaced by “osteoarthritis”. This should be handled consistently throughout the manuscript.
- corrected
Main text body
- I would rearrange chapter 2 and 1. Chapter 2 “Osteoarthritis” needs to be the first chapter as it introduces the disease in general. “Inflammation- role of pathogenesis of OA” should be then chapter 2. I guess it was a misinterpreting of my comments to place the introducing chapter about "osteoarthritis" as chapter 2.
- We modified the manuscript and we started the review with a chapter: Osteoarthritis - To this end, there is a misnumbering of chapters. “Materials and methods” should be chapter 8 and not chapter 9! Consequently, chapter 10 –conclusions- needs to be then chapter 9. - corrected
Another round of proof reading was conducted by native speaker editing services.
Reviewer 2 Report
I feel that the content was organized by narrowing down the target disease to osteoarthritis.
There are many reviews on the relationship between platelets and rheumatoid arthritis, but I feel there are comparatively few on the relationship between platelets and osteoarthritis. So, this review is valuable.
Unfortunately, the points I pointed out last time have not been improved.
In a newly inserted paragraph on page 5, the original terms of the abbreviations explained on page 2 are listed again.(Interleukin, transforming growth factor, Tumor Necrosis Factor [on page 6])
I would like the author to organize the whole thing and read through the document. Since this is a review, please be more careful in your use of terminology.
Author Response
First of all, thank you for your comments and suggestions which allowed us to greatly improve the quality of the article. We agree with all your comments, and we modified the manuscript point by point accordingly.
In a newly inserted paragraph on page 5, the original terms of the abbreviations explained on page 2 are listed again.(Interleukin, transforming growth factor, Tumor Necrosis Factor [on page 6])
I would like the author to organize the whole thing and read through the document. Since this is a review, please be more careful in your use of terminology.
- Article was overseen and modified accordingly to your suggestions. We detected no more mistakes with the original terms and the abbreviations.
Another round of proof reading was conducted by native speaker editing services.